# The *N125S* polymorphism in the cathepsin G gene *(rs45567233)* is associated with susceptibility to osteomyelitis in a Spanish population

Laura Pérez-Is[1,2], Marcos G. Ocaña[3], A. Hugo Montes[1,2], José A. Carton[2,4], Victoria Álvarez[5], Álvaro Meana[6], Joshua Fierer[7], Eulalia Valle-Garay[1,2], Víctor Asensi [2,4] *

1 Biochemistry and Molecular Biology, University Medical School, Oviedo, Spain, 2 Group of Translational Research in Infectious Diseases, Instituto de Investigación Sanitaria del Principado de Asturias (ISPA), Oviedo, Spain, 3 Biotechnological and Biomedical Assays Unit, University Medical School Oviedo, Spain, 4 Infectious Diseases, Hospital Universitario Central de Asturias, University Medical School, Oviedo, Spain, 5 Molecular Genetics Unit-Nephrology Research Institute, Hospital Universitario Central de Asturias, Oviedo, Spain, 6 Community Center for Blood and Tissues of Asturias, CIBERER U714, Oviedo, Spain, 7 Infectious Diseases Section, VAMC and University of California, San Diego, California, United States of America

* vasensia@gmail.com

**Data Availability Statement:** All data are contained within the paper.

## Abstract

### Background

Osteomyelitis is a bone infection, most often caused by *Staphylococcus aureus*, in which neutrophils play a key role. Cathepsin G (CTSG) is a bactericidal serine protease stored in the neutrophil azurophilic granules. CTSG regulates inflammation, activating matrix metallo-proteinases (MMPs), and coagulation. Lactoferrin (LF), a neutrophil glycoprotein, increases CTSG catalytic activity and induces inflammation. The aim of this study was to analyze a potential association between a *CTSG* gene polymorphism (*Asn125Se*r or *N125S*, *rs45567233*), that modifies CTSG activity, and could affect susceptibility to, or outcome of, bacterial osteomyelitis.

### Methods

*CTSG N125S* polymorphism was genotyped in 329 osteomyelitis patients and 415 controls), Blood coagulation parameters, serum *CTSG* activity, LF, MMP-1, MMP-13, and soluble receptor activator for nuclear factor κ B ligand (sRANKL) levels were assessed in carriers of the different *CTSG* genotypes.

### Results

*CTSG N125S* (*AG*) genotype was significantly more frequent among osteomyelitis patients than controls (15.5% vs. 9.4%, p = 0.014). *CTSG N125S* variant *G* allele (*AG +GG*) was also more frequent among osteomyelitis patients (8.1% vs. 4.7%, p = 0.01). Serum CTSG activity and LF levels were significantly higher in osteomyelitis patients carrying the *G* allele

**Funding:** This study was partially funded by a Fondo de Investigaciones Sanitarias (FIS) grant PI16/01999 given to VA. The funder had no role in study design, data collection and analysis, decision to publish, or preparation of the manuscript. There was no additional external funding received for this study.

**Competing interests:** The authors have declared that no competing interests exist.

compared to those with the *AA* genotype, ($p<0.04$). Serum MMP-1 was lower in the *G* allele carriers ($p = 0.01$). There was no association between these genotypes and clinical characteristics of osteomyelitis, or coagulation parameters, MMP-13, and sRANKL serum levels.

## Conclusions

Differences in the *CTSG* gene might enhance osteomyelitis susceptibility by increasing CTSG activity and LF levels.

## Introduction

Cathepsin G (CTSG) is a 26-kDa serine protease stored in the azurophilic granules of the polymorphonuclear leukocytes. It is released as a consequence of neutrophil stimulation by platelet-activating factor and different cytokines, and CTSG then enhances platelet aggregation. CTSG has antimicrobial activity against several different species of bacteria. It also plays an important role on the breakdown of extracellular matrix (ECM) components by activating different extracellular matrix metalloproteinases (MMPs) [1–3]. CTSG also stimulates the production of receptor activator for nuclear factor κ B ligand (RANKL), critical in bone remodeling, by virtue of activating osteoclast precursors [4]. CTSG has been reported to play an important role in a variety of inflammatory diseases including rheumatoid arthritis, coronary artery diseases, and ischemic reperfusion injury, and also response to bone metastasis. It is also implicated in several dental and respiratory infectious and inflammatory diseases, including periodontitis, chronic obstructive pulmonary disease, acute respiratory distress syndrome, and cystic fibrosis [1, 2].

The *CTSG* gene is located in chromosome 14q11.2, spans 2.7 kb and consists of 5 exons and 4 introns. An A→G missense mutation in exon 4 of the *CTSG* gene (*Asn125Se*r or *N125S*, *rs45567233*) was reported [5]. This point mutation at position 2108 changes the codon *AAC* ($^{125}$Asn) to *AGC* ($^{125}$Ser). Single carriers of the variant *G* genotype are heterozygous (*GA*), double carriers are homozygous (*GG*), while those that do not carry the variant *G* allele are wild type and have the *AA* genotype. The *N125S* polymorphism, that might modify CTSG activity, associates with elevated plasma fibrinogen levels in addition to increased platelets activation, contributing to cardiovascular and cerebrovascular disease [6]. No associations of the *CTSG* *N125S* polymorphism with any infections have been reported [7].

Lactoferrin (LF), a glycoprotein also stored in neutrophil granules, and it increases CTSG catalytic activity at physiological concentrations. LF also enhances CTSG-induced platelets activation and inflammatory mediators, modulating the inflammatory response [8, 9]. Interestingly high LF parotid saliva levels have been observed in patients with periodontitis, an inflammatory condition that can progress to mandibular osteomyelitis [9].

Osteomyelitis is a bacterial bone infection characterized by progressive bone inflammatory destruction and necrosis sequestra and new bone formation (involucrum). Adult osteomyelitis is frequently a complication of compound fractures, and/or open bone surgery. This infection can also develop from contiguous spread of infection from open wounds such as chronic pressure ulcers. Hematogenous osteomyelitis occurs mostly in prepuberal children. In adults, hematogenous osteomyelitis usually affects the axial skeleton. *Staphylococcus aureus* is the microorganism most frequently isolated in both post-traumatic and hematogenous osteomyelitis [10]. In spite of its frequency, complicated therapy and frequent relapses, osteomyelitis pathogenesis has been scarcely studied. Neutrophils, one of the main host defenses against

bacterial infections, play a key role in osteomyelitis pathogenesis. Our group has previously reported that there are changes in the life span of neutrophils of osteomyelitis patients [11]. CTSG *N125S* polymorphism, a genetic variation that might modify neutrophils CTSG activity could influence osteomyelitis pathogenesis by modifying LF production and subsequent cleavage, MMPs-mediated bone inflammation and remodeling, and activating coagulation.

The aim of this study was to analyze a potential association between bacterial osteomyelitis and the *CTSG N125S* polymorphism, and the effect of that genetic polymorphism on CTSG activity. We also assessed associations with potential downstream consequences of CTSG release on some aspects of coagulation, and levels of circulating LF, MMP -1 and -13 and the soluble receptor activator for nuclear factor κ B ligand (sRANKL).

## Patients and methods

### Patients

We enrolled 329 adult patients admitted to the Hospital Universitario Central de Asturias (HUCA) and three affiliated hospitals in the same Northern Spanish region with a diagnosis of bacterial osteomyelitis, between January 1998 and December 2018. These hospitals provide the health coverage to the region of Asturias (total population of 1 million people). Patients with acute and chronic osteomyelitis were included in the study and followed for one year. Clinical information regarding these patients has been published in detail [12]. A diagnosis of osteomyelitis was made by clinical and roentgenographic criteria [10]. Surgical and sinus tract pus samples were cultured in all the osteomyelitis patients to make an etiological diagnosis. The presence of bone sequestra and/or sinus tract in bone X ray, CT or MRI, a positive $Ga^{67}$ uptake bone scan, or a positive culture of the surgical bone sequestra or sinus tract were diagnostic of osteomyelitis [10]. Patients were diagnosed with hematogenous osteomyelitis if their bone infection was acquired in absence of trauma or bone surgery, lower limb vascular insufficiency, or a contiguous focus of infection. Overall, 56 cases of hematogenous osteomyelitis, 27 (48.2%), with positive blood cultures, were included in the study Bacterial osteomyelitis in immunosuppressed patients undergoing chemotherapy or with bone tuberculosis were excluded. Osteomyelitis that was present for more than 3 months was considered as chronic and if it did not relapse within a year of follow-up was considered as "cured". The DNA from all enrolled patients was stored at -20° C and has been used previously for several gene association studies [13–19]. In addition, 415 healthy HUCA Blood Bank donors, matched for sex and age with the osteomyelitis patients, were used as controls. Patients and controls were members of a homogeneous Caucasian population, and were residents of Asturias, a Northwestern Spanish region with a small foreign immigrant population (less than 5%). Each participant gave informed written consent for the study, which was approved by the Ethical Committee of Regional Clinical Research of the Principality of Asturias.

### Blood coagulation parameters

All the osteomyelitis patients enrolled in the study had 20 ml of blood drawn at hospital admission. Ten ml were used for hemogram, standard biochemistry, and coagulation assays. Routine coagulation parameters included fibrinogen levels, prothrombin time (PT), and international normalized ratio (INR), and activated partial thromboplastin time (aPTT).

### Measurement of the *N125S (rs45567233) g*enetic polymorphism of *CTSG* gen

Ten ml of blood from each patient and controls were collected in potassium-EDTA glass tubes. Genomic DNA was extracted from peripheral leukocytes with a salting-out method and

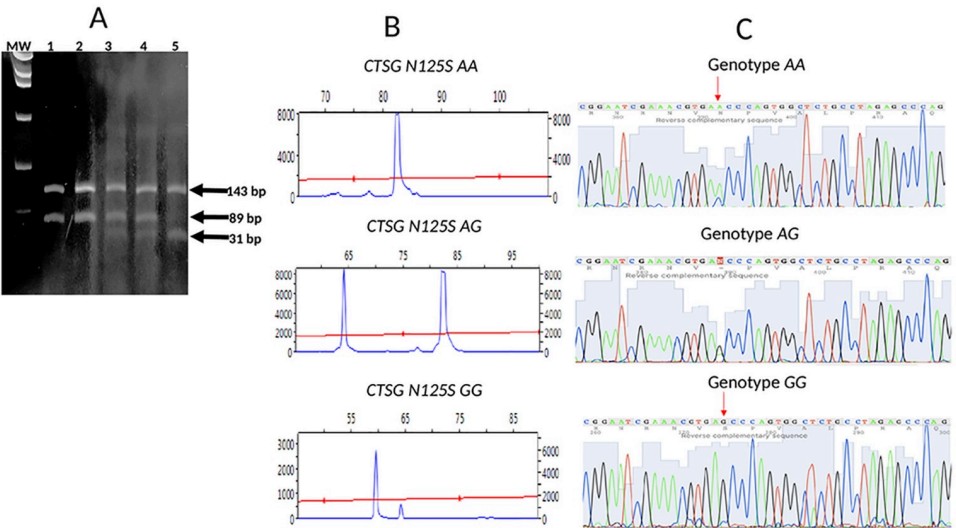

**Fig 1. Detection of the *N125S (rs 45567233)* polymorphism in the *CTSG* gene.** (A): polyacrylamide gel analysis of genomic DNA from 5 patients with acute osteomyelitis (OM) processed as in methods. *AA* denotes a wild-type sample with two distinct bands of 143 and 89 bp (lanes 1 and 2); *AG* denotes a heterozygous sample showing three distinct bands of 143, 89 and 31 bp (lanes 3 and 4); *GG* denotes a double variant sample showing two distinct bands of 143 and 31 bp (lane 5). MW, molecular weight markers; (B) Peak scan analysis of 3 patients with the genotypes *AA*, *AG* and *GG*; (C) Sequencing of the restriction fragment length polymorphism. Samples with no alteration: wild type (upper); with a heterozygous *A-to-G* replacement (middle) and with a double *A-to-G* replacement (lower).

stored at -20˚ C until use. The *N125S* (*rs45567233)* polymorphism of the *CTSG* was analyzed in patients and controls by PCR. The PCR primers used for amplifying the region of the gene containing the *N125S* polymorphism (*rs45567233*) were: Forward: `5'-6-FAM-GCTGAGC GGGAACGCCTACA-3'` and Reverse: `5´-GCTGAGCGGGAACGCCTACA-3'` [6–7]. These primers amplified a fragment of 263 bp. The amplification protocol consists on an initial denaturation at 94˚ C for 5 minutes, followed by 35 cycles of denaturation at 94˚ C for 1 minute, annealing at 61˚ C for 1 minute and elongation at 72˚ C for 1 minute, and a final extension at 72˚ C for 7 minutes. PCR products were incubated overnight at 37˚ C with the restriction enzyme SduI (Thermo Fisher Scientific Inc, Waltham, MA, USA) followed by capillary electrophoresis on ABI PRISM® 3130xl Genetic Analyzer (Applied Biosystem, Foster City, CA, USA). Results were assessed using the Peak Scanner™ Software v1.0 (Applied Biosystem, Foster City, CA, USA). The *CTSG N125S* polymorphism was also genotyped individually for each osteomyelitis patient and control. PCR products were electrophoresed on a 2% low-melting agarose gel, and the fragments were then excised from the gel, purified with spin columns (DNA gel extraction Kit; Millipore, Billerica, MA,USA), and the fragments were directly sequenced on an ABIPrism 310 Genetic Analyser (Applied Biosystems, Foster City, CA, USA).(Fig 1).

## CTSG, LF, MMPs and sRANKL assessment

Serum from patients with acute osteomyelitis was used for CTSG, LF, MMPs and sRANKL assessment. All the patients used in CTSG, LF, MMPs and sRANKL assessment assays had been consecutively admitted to the HUCA between 2016–18.

## Serum CTSG activity

Cathepsin G activity was determined in serum of 21 patients with acute osteomyelitis and 21 blood donors were used as controls by a commercially available colorimetric assay (Cathepsin

G Activity Assay Kit, PK-CA577-K146, PromoCell GmbH, Heidelberg, Germany). CTSG activity levels were expressed as μU/ml.

### Serum Lactoferrin (LF)

LF was determined in serum of 27 patients with acute osteomyelitis by an ELISA assay (ab200015 human lactoferrin simpleStep ELISA Kit Abcam, Cambridge, MA, USA). LF levels were expressed as ng/ml.

### Serum MMPs and sRANKL

MMP-1 and -13 and sRANKL were determined in 21 patients with acute osteomyelitis by ELISA and by commercial available colorimetric assays (Raybiotech, Norcross, GA, USA for MMP-1 and 13, Immunodiagnostics, Bensheim, Gerrmany for sRANKL). Serum MMPs levels are expressed in pg/ml.and sRANKL levels in pmol/L.

## Statistics

### Sample size calculation

Power calculation using one control for each osteomyelitis patients indicated a need for 746 individuals, 373 osteomyelitis patients and 373 controls to have a power of 80% at a confidence interval of 90% to detect a prevalence of 17% of the *CTSG N125S* polymorphism carriage among osteomyelitis patients considering that this polymorphism has a known prevalence of 10% in the Caucasian European population.

Results are expressed as median and inter-quartile range (IQR) or proportions as appropriate. As the distribution of some continuous variables was non-Gaussian, natural logarithmic transformation was done for analysis. The reported values are the result of back-transformation into the original units. The Pearson $X^2$ test was used to compare allele and genotype frequencies between the groups. Yates' correction and Mantel–Haenszel test were also used when indicated. Odds ratios (OR´s) and their 95% confidence intervals (CI) were also calculated. Serum CTSG, LF, MMP-1, -3, sRANKL were compared for the genotypes with the Student t test or the Mann-Whitney test when appropriate. Serum CTSG and LF, were compared by using the Pearson correlation coefficient. All the reported p values are two-sided. A p-value < 0.05 was considered as significant. The statistical analysis was performed with the computer program statistical package SPSS (IBM SPSS Statistics 25.0 package, IBM New York, NY, USA) and with the GraphPad Prism software (GraphPad Software, version 7.0, San Diego, CA, USA).

## Results

### Clinical characteristics of osteomyelits patients

Osteomyelitis patients were mainly men over sixty, with infected bones after fractures (38.9%). Pressure ulcers (23.4%), prosthetic joint infection (20.7%) and hematogenous spread (17.2%) were the risk factors for about half the cases of osteomyelitis. The most commonly involved bones were tibia (24.1%) and femur (21.8%). The most common pathogen was *S. aureus* and most of the bone infections were chronic (Table 1). The controls were blood donors mentioned in methods; 295 (71.1%) were males and 120 (28.9%) were females with a median age of 59.1 years (41–60). Although the controls were on average 2.9 years younger than the patients, there was not a significant difference between the ages of the patients and controls (p = 0.14).

**Table 1. Clinical characteristics of the 329 osteomyelitis (OM) patients enrolled in the study.**

| Clinical Characteristics | Patients (n = 329) |
|---|---|
| Median age (years, range) | 62.0 (49–74) |
| Male gender (n, %) | 243 (73.9) |
| Chronic OM (n, %) | 215 (65.3) |
| Post-traumatic source of infection (n, %) | 128 (38.9) |
| Hematogenous source of infection (n, %) | 56 (17.2) |
| Paraplegia/pressure ulcers infection (n, %) | 77 (23.4) |
| Orthopedic prosthesis infection (n, %) | 68 (20.7) |
| *Staphylococcus aureus* OM (n, %) | 139 (67.5) * |
| Gram negative OM (n, %) | 48 (23.8) * |
| Other microorganisms (n, %) | 19 (23.4) * |

* Positive cultures were available only in 206 osteomyelitis patients

## Frequency of the *CTSG N125S (rs 45567233)* polymorphism in osteomyelitis

Tables 2 and 3 show the genotypic and allelic frequencies of the *CTSG N125S (rs 45567233)* polymorphism in osteomyelitis patients and controls. The *CTSG N125S AG* heterozygous genotype was significantly more frequent among osteomyelitis patients compared to controls (15.5% vs. 9.4%; $\chi^2 = 6.5$, OR = 1.78, 95% CI = 1.11–2.84, p = 0.011 by the Mantel-Haenszel test, $\chi^2 = 5.95$, p = 0.014 by the Yates correction). The *CTSG N125S* variant *G* allele was also more frequent among osteomyelitis patients (8.1% vs 4.7%, $\chi^2 = 7.12$, OR = 1.78, 95% CI = 1.14–2.78, p = 0.0076 by the Mantel-Haenszel test, $\chi^2 = 6.56$, P = 0.01 by the Yates correction). Only one carrier of the double variant *GG* genotype was found among osteomyelitis patients (0.3%) and none among the controls. The *CTSG N125S* polymorphism was in Hardy–Weinberg equilibrium among patients with osteomyelitis and controls. No association of the

**Table 2. Frequency of CTSG N125S (rs 45567233) genotypes in osteomyelitis (OM) patients and blood donor (controls).**

| *CTSG N125S* Genotypes | OM patients (n, %) | Controls (n, %) |
|---|---|---|
| *AA* | 277 (84.2) | 376 (90.6) |
| *AG* | 51 (15.5) * | 39 (9.4) |
| *GG* | 1 (0.3) | 0 (0) |

* p = 0.011 by the Mantel-Haenszel test; p = 0.015 by the Yates's correction while comparing OM patients vs. controls.

**Table 3. Frequency of CTSG N125S (rs 45567233) alleles in osteomyelitis (OM) patients and blood donor controls.**

| *CTSG* Alleles | OM patients (n, %) | Controls (n, %) |
|---|---|---|
| *A* | 605 (91.9) | 791 (95.3) |
| *G* | 53 (8.1) * | 39 (4.7) |

* p = 0.008 by the Mantel-Haenszel test, p = 0.01 by the Yates correction comparing OM patients vs. controls.

*CTSG N125S* polymorphism with type of bone infection (acute vs. chronic) (p = 0.7), source of infection (hematogenous vs. non hematogenous, p = 0.9), (pressure ulcers vs. non pressure ulcers, p = 0.5) or microorganism isolated (*S. aureus* vs. Gram negative bacteria p = 0.9) was found.

### Serum CTSG activity and *CTSG N125S (rs 45567233)* polymorphism

Serum CTSG activity was significantly higher in carriers of the *G* allele of the *N125S* polymorphism compared to those with the *AA* genotype (p = 0.016) and to uninfected controls (p = 0.005) (Fig 2). No significant differences in serum CTSG activity between osteomyelitis patients and controls were observed (p = 0.1).

### Serum LF and *CTSG N125S (rs 45567233)* polymorphism

Serum LF was significantly higher in carriers of the *G* allele of *CTNG* compared to those with the *AA* genotype (p = 0.006) (Fig 3). On the other hand, LF levels and CTSG activity were correlated but the association was just below the limit of significance (r = 0.501, p = 0.057 by Pearson correlation coefficient).

### Serum MMP-1, -13 and sRANKL levels and *CTSG N125S (rs 45567233)* polymorphism

Serum MMP-1 levels were significantly lower in osteomyelitis patients with acute infection who were carriers of the *G* allele of the *N125S* polymorphism compared to those with the *AA* genotype (p = 0.024) (Fig 4). We found no differences in MMP-13 or sRANKL serum levels between osteomyelitis patients who were carriers of the different alleles of *CTSG 125* (124.3 [8.6–240.0] and 89 [55.05–304.0] pg/ml, p = 0.9 for MMP-13, and 0.165 [0.04–0.250] and 0.08 [0.04–0.210] pmol/L, p = 0.7 for sRANKL for osteomyelitis patients carriers of the *AA* and *AG* + *GG* genotypes of the *CTSG*).

### Coagulation parameters and *CTSG N125S (rs 45567233)* polymorphism

There were no differences in blood fibrinogen levels, PT, or PTT between the osteomyelitis patients who carried the *CTSG* and those that did not (median [IQR] 621[433–772] and 622 [492.3–783.8] mg/dl, for the *AG* and *AA* genotypes, respectively [p>0.6]), (PT: 11.7 [110–12.6] and 11.6 [10.7–12.5] seconds, p = 0.1; INR: 1.1 [1.0–1.2] and 1.1 [1.0–1.1], p = 0.3; aPTT: 31.4 [28.8–33.8] and 32.0[28.4–33.8] seconds, p = 0.4, for the *AG* and *AA* genotypes, respectively).

## Discussion

We describe for the first time an association in adults between the *CTSG N125S* polymorphism, previously associated with cardiovascular and neurovascular diseases, and osteomyelitis, a bacterial infection of the bone. This *CTSG* polymorphism is associated with increased serum CTSG activity and LF levels in our study. Carriers of the *G* allele were over-represented in patients with osteomyelitis compared to a control population drawn from the same area of Spain where the patients were hospitalized, suggesting that this allele increases the risk of developing osteomyelitis. The carriage of this variant *G* allele is relatively frequent (9.4%) in the healthy Spanish Caucasian controls in this study, but was even higher (15.5%) in our patients with osteomyelitis. We have previously found that variants in the genes of *IL-1α*, *MMP1*, *TLR4* receptor, endothelial nitric oxide synthase (*NOS3*), *Bax* and *tPA* also contribute to the risk of developing osteomyelitis with prevalences of between 3.8% and 65.3% [14–19]. The *CTSG N125S* polymorphism frequency in our control population was very similar to that

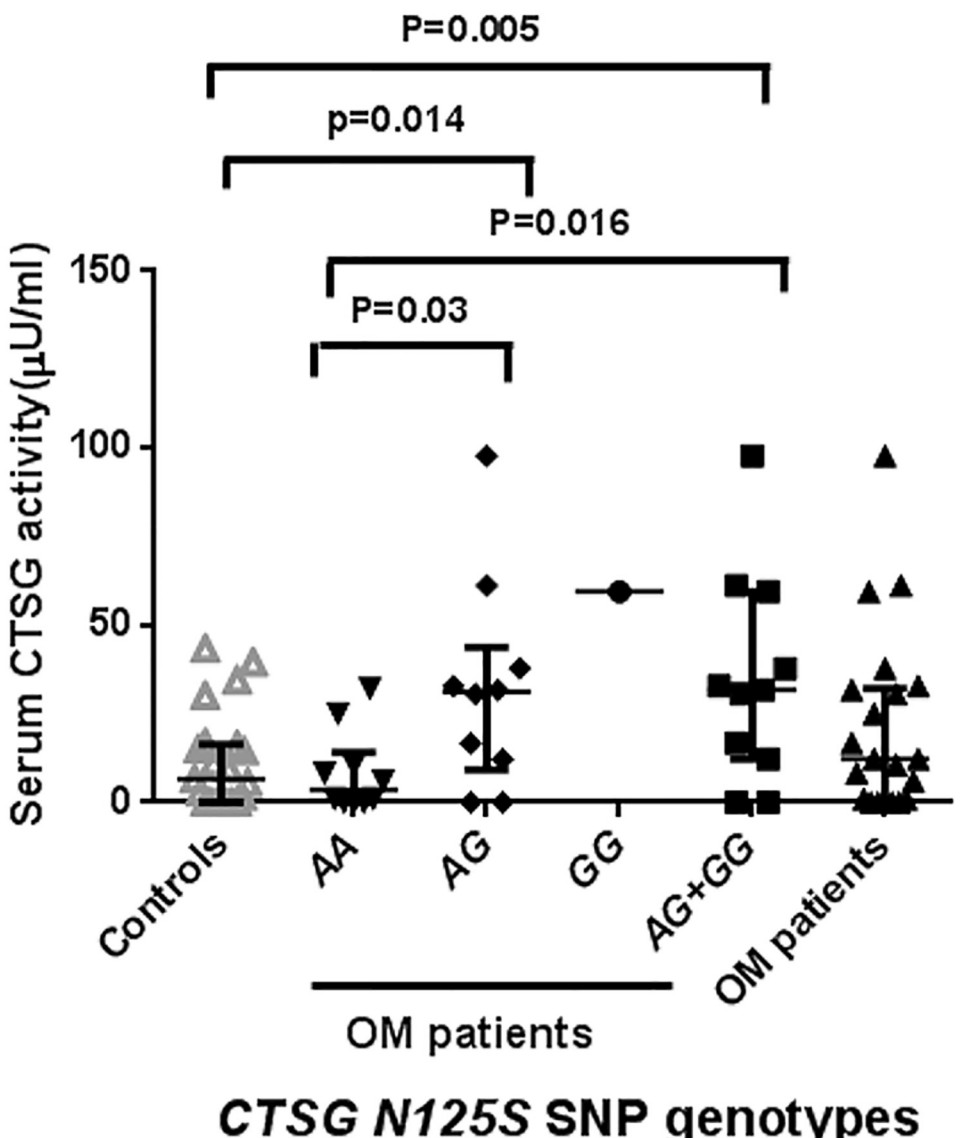

**Fig 2. Serum cathepsin G (CTSG) activity in carriers of the different genotypes of the *CTSG N125S (rs 45567233)* polymorphism.** Cathepsin G (CTSG) activity was determined in serum of osteomyelitis patients and controls by a colorimetric assay. Data are shown as the median and IQR of 21 patients with acute osteomyelitis, 10 carriers of the *AA*, 10 of the *AG* and 1 of the *GG* genotypes of the *CTSG N125S (rs 45567233)* polymorphism and 10 blood donors were assessed.

reported by others (10% in healthy Caucasian Americans and 11.7% in healthy French and British Caucasians) [6, 20]. This encourages us to consider that our finding was not due to a peculiarity of the population we studied.

Previous reports did not find increased *in vitro* transcriptional activities due to the *CTSG N125S* polymorphism in human monocyte-like U937 cells, a cell-line that expresses 25% more CTSG activity than human neutrophils [6, 21, 22]. We report here for the first time that the *CTSG N125S* polymorphism is associated with increased plasma CTSG activity in carriers of the variant *G* allele. This increased activity was due to the effect of the *G* allele of

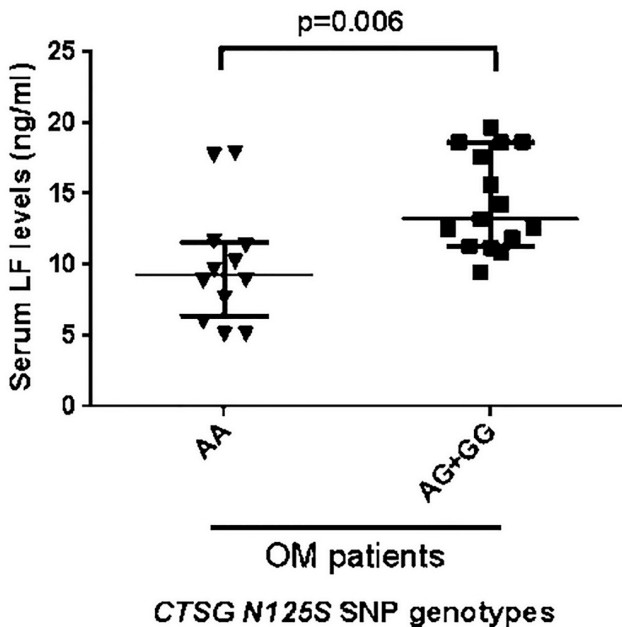

**Fig 3. Serum Lactoferrrin (LF) levels in patients with acute osteomyelitis according to their alleles of *CTSG (rs 45567233)*.** Lactoferrin (LF) was measured in serum of osteomyelitis patients by an ELISA assay. Results from individual *AA* subjects are shown as inverted triangles and *AG + GG* subjects as squares. The long horizontal bars are the median values and shorter bars show IQR.

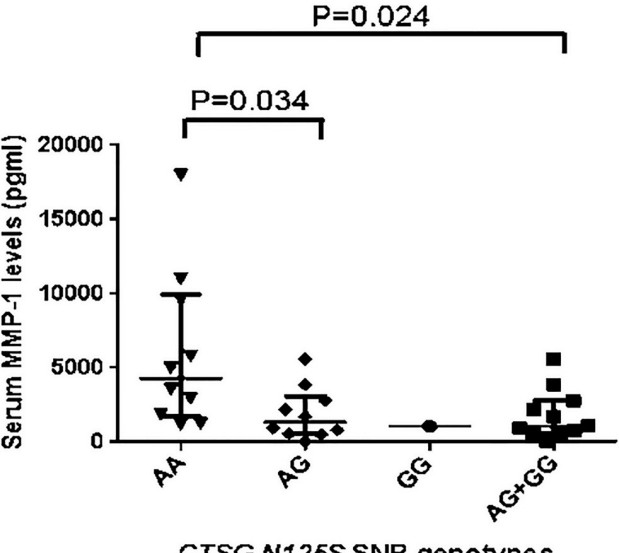

**Fig 4. Serum MMP-1 levels in carriers of the different genotypes of the *CTSG (rs 45567233)*.** Serum MMP-1 levels from 27 patients with acute osteomyelitis were determined after the end of treatment by ELISA and the group was divided according to whether or not they carried the *G* allele. Individual patients' results are shown and the median and IQR. There were 10 carriers of the *AA*, 10 of the *AG* and 1 of the *GG* genotypes of the *CTSG N125S (rs 45567233)* polymorphism.

the *CTSG N125S* polymorphism and not to the bone infection as it also was true of the uninfected controls.

We also observed that serum LF levels were significant higher in carriers of the variant *G* allele of the *CTSG N125S* polymorphism. These individuals also had increased serum CTSG serum activity. Considering that LF increases the catalytic activity of CTSG at physiological concentration and broadens the substrate specificity of CTSG [8], our LF levels finding could partially explain the increased activity of serum CTSG. However, there was not a statistically significant correlation between serum CTSG activity and LF levels, which could mean that the N→S change in CSTG also changed its enzymatic activity.

We also explored the mechanisms whereby the *CTSG N125S* polymorphism might increase susceptibility to osteomyelitis. CTSG is released after neutrophil stimulation by platelet-activating factor, tumor necrosis factor-alpha, and interleukin-8 which, via the CTSG platelet receptor protease -activated receptor 4, leads to calcium mobilization, platelet secretion and aggregation and a systemic release of the platelet thrombogenic products producing intravascular thrombosis [3, 6]. The *CTSG N125S G* allele has been associated with elevated plasma fibrinogen levels in myocardial infarct patients [6, 23]. Prothrombotic polymorphisms such as the *tPA Alu (I/D)* have been associated with osteomyelitis [19]. However, fibrinogen levels and other blood coagulation parameters were similar in carriers of the different genotypes of the *CTSG N125S* polymorphism among osteomyelitis patients in our study.

CTSG plays a role in the inflammatory response and it has been associated with different inflammatory diseases, especially some related to the bone such as rheumatoid arthritis, periodontitis, and bone metastasis. CTSG degrades collagen and proteoglycan and digests ECM components at inflammatory sites to which neutrophils are recruited by activating MMP-1, -2, -3 and RANKL [4, 24–26]. CTSG increases MMP expression in normal human fibroblasts through fibronectin fragmentation, and induces the conversion of pro-MMP-1 to active MMP-1 [24]. CTSG is also capable of activating pro-MMP-9, that cleaves and releases active transforming growth factor-β (TGFβ), MMP-13 and RANKL at the tumor-bone interface of mammary tumor-induced osteolytic lesions [26]. These MMPs and RANKL interfere in the bone remodelling by enhancing osteoclasts activation and bone resorption, potentially aggravating the bone damage already induced by infection in osteomyelitis. We have reported increased plasma levels of MMP-1 and MMP-13 in osteomyelitis patients in a previous work [18]. However, although MMP-1 plasma levels were increased in osteomyelitis *AA* carriers, MMP-1 plasma levels were not increased but were significantly decreased in the osteomyelitis patients with the *CTSG N125S* allele in this study. There were no differences in plasma levels of MMP-13 and RANKL among osteomyelitis patients with the different genotypes of the *CTSG*. Therefore, mechanisms other than increased MMP-1, -13 and sRANKL might explain the increased susceptibility to osteomyelitis in carriers of the *CTSG N125S* polymorphism.

The CTSG *N125S* polymorphism, increased LF serum and probably bone levels as well As an inducer of inflammation, LF obtained from human parotid saliva increased the production of IL-6, monocyte chemoattractant protein-1 (MCP-1) and the activation of NF-κB in human epithelial HSC-2 cells [9]. Perhaps the association of the CTSG *N125S* polymorphism with osteomyelitis might be mediated by LF. However Komine et al found that the activity was in fragments of LF not in the whole protein and the response was measured with an epithelial cell line that might not be pertinent to bones.

However, there are still other plausible explanations of the association of the CTSG polymorphism with osteomyelitis. Neutrophil CTSG is important during the early inflammation stage of wound healing. CTSG may be involved in processing different soluble mediators in the wound milieu that are responsible for neutrophil chemotaxis. *CTSG* KO mice have abnormalities in wound healing manifested by a decrease in wound tensile strength in spite of

having an increased number of neutrophils and more myeloperoxidase activity in the wound [27]. In spite of the increased serum CTSG activity associated with the *CTSG* polymorphism we could speculate that there was abnormal wound healing that could play some role in the enhanced susceptibility to osteomyelitis. Many of the patients contracted bone infection as a complication of unhealed open wounds after orthopedic surgery or from contiguous spread to bone from chronic pressure ulcers in patients with spinal cord lesions. However, no association between osteomyelitis due chronic pressure ulcers and the *CTSG N125S* polymorphism was found in our study. Finally, a linkage disequilibrium of the *CTSG* gene with another gene located on chromosome 14q11-2 cannot be ruled-out.

In summary, we describe here a potential association between the *CTSG N125S* polymorphism and osteomyelitis. The effect might be due to bone inflammation enhanced by increased LF local levels. More research in populations with other ethnic background is needed to confirm this observation and to discover the mechanism underneath the increased susceptibility to bone infection due to this genetic variant.

## Acknowledgments

The results were presented in part at the 2017 ASM Microbe Conference, New Orleans, Louisiana, June 1 to 5, abstract # 855.

## Author Contributions

**Conceptualization:** Marcos G. Ocaña, José A. Carton, Victoria Álvarez, Eulalia Valle-Garay, Víctor Asensi.

**Data curation:** Laura Pérez-Is, Marcos G. Ocaña, José A. Carton, Víctor Asensi.

**Formal analysis:** A. Hugo Montes, Víctor Asensi.

**Funding acquisition:** Víctor Asensi.

**Investigation:** Laura Pérez-Is, Marcos G. Ocaña, A. Hugo Montes, José A. Carton, Álvaro Meana, Eulalia Valle-Garay, Víctor Asensi.

**Methodology:** Laura Pérez-Is, Marcos G. Ocaña, A. Hugo Montes, Álvaro Meana, Eulalia Valle-Garay.

**Project administration:** Eulalia Valle-Garay, Víctor Asensi.

**Resources:** Victoria Álvarez, Álvaro Meana, Eulalia Valle-Garay.

**Software:** A. Hugo Montes.

**Supervision:** Victoria Álvarez, Eulalia Valle-Garay, Víctor Asensi.

**Validation:** A. Hugo Montes, Victoria Álvarez, Eulalia Valle-Garay, Víctor Asensi.

**Visualization:** A. Hugo Montes, Víctor Asensi.

**Writing – original draft:** Víctor Asensi.

**Writing – review & editing:** José A. Carton, Joshua Fierer, Eulalia Valle-Garay, Víctor Asensi.

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
