## [Decision Letter · Decision Letter 0]

23 Aug 2019

PONE-D-19-18899

The N125S polymorphism in the G gene is associated with susceptibility to<gwmw class="ginger-module-highlighter-mistake-type-3" id="gwmw-15664826668904541061438"> .</gwmw>

PLOS ONE

Dear Dr. Asensi,

Thank you for submitting your manuscript to PLOS ONE. After careful consideration, we feel that it has merit but does not fully meet PLOS ONE’s publication criteria as it currently stands. Therefore, we invite you to submit a revised version of the manuscript that addresses the points raised during the review process.

We would appreciate receiving your revised manuscript by Oct 07 2019 11:59PM. To enhance the reproducibility of your results, we recommend that if deposit your laboratory protocols in protocols.io, where a protocol can be assigned its own identifier (DOI) such that it can be cited independently in the future. For<gwmw class="ginger-module-highlighter-mistake-type-3" id="gwmw-15664826752933044759050"> :</gwmw> http://journals.plos.org/plosone/s/submission-guidelines#loc-laboratory-protocols

A rebuttal letter that responds to each point raised by the academic editor and reviewer(s). This letter should be uploaded as and labeled 'Response to Reviewers'.A marked-up copy of your manuscript that highlights changes made to the original version. This file should be uploaded as and labeled 'Revised Manuscript with Track Changes'.An unmarked version of your revised paper without changes. This file should be uploaded as and labeled 'Manuscript'.

<gwmw class="ginger-module-highlighter-mistake-type-2" id="gwmw-15664826912533761596689">Please</gwmw> forming your response, if your article is accepted, you may have the opportunity to make the peer review history publicly available. The record will include editor decision letters (with reviews) and your responses to reviewer comments. If eligible, we will contact you to opt in or out.

We look forward to receiving your revised manuscript.

Kind regards,

Manal S. <gwmw class="ginger-module-highlighter-mistake-type-1" id="gwmw-15664832293081786252396">Fawzy</gwmw>, Ph.D., M.D.

Academic Editor

PLOS ONE

Journal Requirements:

2. Please provide additional details regarding participant consent. In the ethics statement in the Methods and online submission information, please ensure that you have specified whether consent was written or verbal/oral. If consent was verbal/oral, please specify: a) whether the ethics committee approved the verbal/oral consent procedure, b) why written consent could not be obtained, and c) how verbal/oral consent was recorded. If your study included minors, please state whether you obtained consent from parents or guardians in these cases. If the need for consent was waived by the ethics committee, please include this information.

3. Please note that all PLOS journals ask authors to adhere to our policies for sharing of data and materials: https://journals.plos.org/plosone/s/data-availability. According to PLOS ONE’s Data Availability policy, we require that the minimal dataset underlying results reported in the submission must be made immediately and freely available at the time of publication. As such, please remove any instances of 'unpublished data' or 'data not shown' in your manuscript and replace these with either the relevant data (in the form of additional figures, tables or descriptive text, as appropriate), a citation to where the data can be found, or remove altogether any statements supported by data not presented in the manuscript."

4. We noticed you have some minor occurrence(s) of overlapping text with the following previous publication(s), which needs to be addressed:

https://en.wikipedia.org/wiki/Cathepsin_G

https://doi.org/10.1093/infdis/jit158

https://doi.org/10.1097/GIM.0b013e318039b23d

https://doi.org/10.1016/j.molimm.2007.10.013

In your revision ensure you cite all your sources (including your own works), and quote or rephrase any duplicated text outside the Methods section. Further consideration is dependent on these concerns being addressed.

5. Thank you for including your ethics statement:

"Each participant gave informed consent for the study, which was approved by the Ethics Committee of the HUCA.".

i) Please amend your current ethics statement to include the full name of the ethics committee/institutional review board(s) that approved your specific study.

ii) Once you have amended this/these statement(s) in the Methods section of the manuscript, please add the same text to the “Ethics Statement” field of the submission form (via “Edit Submission”).

6. Thank you for stating in your Funding Statement:

"Partially funded by a Fondo de Investigaciones Sanitarias (FIS) grant PI16/01999 given

to VA.

The funders had no role in study design, data collection and analysis, decision to

publish, or preparation of the manuscript".

7. We note that you have included the phrase “data not shown” in your manuscript. Unfortunately, this does not meet our data sharing requirements. PLOS does not permit references to inaccessible data. We require that authors provide all relevant data within the paper, Supporting Information files, or in an acceptable, public repository. Please add a citation to support this phrase or upload the data that corresponds with these findings to a stable repository (such as Figshare or Dryad) and provide and URLs, DOIs, or accession numbers that may be used to access these data. Or, if the data are not a core part of the research being presented in your study, we ask that you remove the phrase that refers to these data.

Additional Editor Comments:

1- The authors should use a language-editing service to refine the use of English in their manuscript and submit an "Editing Certificate" with the revised version of the manuscript.

2- Abbreviations must be spelled out on first mention particularly in the abstract.

3- Authors are advised to revise all the manuscript to ensure that all gene names are written in italic font.

4- Authors should pay attention to the use of "Gender" terminology and substitute by the "sex" terminology as it appears from the results they mean the normal biological differences between males and females.

5- Thanks to the authors for providing the PCR primer sequences. Are these primers -designed or derived from other published work? If the former, please provide the name of the program you applied in its formal citation in the text, or provide the citations you follow if they were derived from other publications.

6- The PCR results need more elaboration and clarification (recommended for acceptance). The authors wrote that "PCR products were incubated overnight at 37º C with the restriction enzyme<gwmw class="ginger-module-highlighter-mistake-type-3" id="gwmw-15664829754791295809559">.</gwmw> Results were determined by using the Peak ScannerTM Software v10". The authors should send a photo related to the original scanner and an edited one for publication (in which all the details of band size for each genotype are written clearly) to facilitate replication of the work by future readers.

7- The quality control measurements the authors followed either in their PCR or other laboratory works, including ELISA, etc. <gwmw class="ginger-module-highlighter-mistake-type-1" id="gwmw-15664831811790536589857">should</gwmw> be written in details.

Reviewers' comments:

Reviewer's Responses to Questions

**Comments to the Author**

1. Is the manuscript technically sound, and do the data support the conclusions?

Reviewer #1: Partly

Reviewer #2: Yes

2. Has the statistical analysis been performed appropriately and rigorously? 

Reviewer #1: Yes

Reviewer #2: Yes

3. Have the authors made all data underlying the findings in their manuscript fully available?

The PLOS Data policy requires authors to make all data underlying the findings described in their manuscript fully available without restriction, with rare exception (please refer to the Data Availability Statement in the manuscript PDF file). The data should be provided as part of the manuscript or its supporting information, or <gwmw class="ginger-module-highlighter-mistake-type-2" id="gwmw-15664829756224531275852">deposited a public</gwmw> repository. For example, in addition to summary statistics, the data points behind means, medians and variance measures should be available. If there are restrictions on publicly sharing data—<gwmw class="ginger-module-highlighter-mistake-type-1" id="gwmw-15664829756322768885661">eg</gwmw>. <gwmw class="ginger-module-highlighter-mistake-type-1" id="gwmw-15664829756364873410917">privacy</gwmw> or use of data from a third party—those must be specified.

Reviewer #1: Yes

Reviewer #2: No

4. Is the manuscript presented in an intelligible fashion and written in standard English?

PLOS ONE does not copyedit accepted manuscripts, so the language in <gwmw class="ginger-module-highlighter-mistake-type-3" id="gwmw-15664829757670599906198">articles</gwmw> must be clear, correct, and unambiguous. Any typographical or grammatical errors should be corrected at revision, so please note any specific errors here.

Reviewer #1: Yes

Reviewer #2: Yes

5. Review Comments to the Author

Please use the space provided to explain your answers to the questions above. You may also include additional comments for the author, including concerns about dual publication, research ethics, or publication ethics. Please upload your review as an attachment if it exceeds 20,000 characters)

Reviewer #1: The study concept is interesting and can be of clinical significance.

My main concern is that the authors recruited an accepted number of patients (329) and controls (415) for and when they made the correlation with the evaluated parameters, they included very small groups of individuals (10 to 27 per group). The validity of these results is an issue and the bias in the selection of these individuals could be another problem.

Other points include

The title should determine the polymorphism ID and the study ethnic group.

-<gwmw class="ginger-module-highlighter-mistake-type-1" id="gwmw-15664829758386099843190">value</gwmw> is sometimes mentioned as "equals" and other occasions as "less than". It should be consistent.

Page 4 last line. The authors mentioned that no similar association studies has been done so far. The reference dated 2006.

Page 5, last paragraph is a mixture of the hypothesis and information extracted from reference no.9. Please rewrite.

There are linguistic style and punctuation errors. The manuscript needs linguistic revision. For example a sentence is repeated page 7, parenthesis <gwmw class="ginger-module-highlighter-mistake-type-3" id="gwmw-15664829758768011961931">in</gwmw> page 19, the tense in discussion (mixture of present and past tenses), first line in the second paragraph p23, .....

The authors provided some data analysis based on the gender. It would be better to remove that as the female group is a minority.

Table 2 is confusing. Please separate into two tables (one for and the other for frequency)

The use of Yates correction is not preferred by some<gwmw class="ginger-module-highlighter-mistake-type-3" id="gwmw-15664829759029781080455"> .</gwmw> I am not sure if it is the best to use here.

Reviewer #2: The authors presented an interesting study evaluating the association between CTSG polymorphisms and the occurrence of and its clinical characteristics. The study is well-conducted, but I suggest the following edits:

* Abstract: The authors did not mention the objective of their study in the abstract.

* Introduction: the authors should present the different alleles, in their study.

* Methods: "We enrolled 329 adult patients with a diagnosis of bacterial between January 1998 and December 2018" Was this a part of a hospital database system or a one study effort that lasted 20 years?

- Also, was there a systematic sample size calculation?

- " <gwmw class="ginger-module-highlighter-mistake-type-1" id="gwmw-15664829759509741182248">was</gwmw> diagnosed using clinical and roentgenographic findings" Please clarify?

- The molecular techniques used in the current study are well-described.

* Results: The age and gender of control subjects and their comparisons to patients should be mentioned in the results' first paragraph. Because they are supposed to be matched, no significant difference should be noted.

- "No association of the CTSG N125S polymorphism with or type of bone infection (acute vs. ), source of infection or microorganism isolated was found" the authors should mention at least p for such associations!

* General: The manuscript needs a professional editing service because there are several grammatical errors across the entire manuscript.

- Data availability statement should be added to the manuscript after the conclusion explaining where the underlying data could be found.

6. PLOS authors have the option to publish the peer review history of their article (what does this mean?). If published, this will include your full peer review and any attached files.

If you choose “no”, your identity will remain your review may still be made public.

Reviewer #1: No

Reviewer #2: No

<gdiv></gdiv><gdiv></gdiv>

---

## [Author Response · Author response to Decision Letter 0]

30 Sep 2019

ANSWERS TO EDITOR AND REVIEWERS 

ANSWERS TO THE EDITOR

Dear Editor,

Below are the answers to all the questions raised by the Editor and Reviewers. Dr. Joshua Fierer, Professor of Medicine and Pathology of the University of California San Diego -UCSD did a complete scientific and English language editing of the original manuscript. This is the reason we decided to include him among the authors of the revised version of the paper. We have introduced a new Fig.1 including gel and peak scan analysis images and RFLP sequences analysis of the N125S (rs 45567233) polymorphism in the CTSG gene from the DNA of 5 osteomyelitis patients. In addition in this revised version of the manuscript we have deleted the S.aureus killing by neutrophils and ROS assays. No differences among genotypes regarding the N125S CTSG polymorphism were observed in these deleted assays and therefore the scientific value of the work has not decreased. Now the manuscript is shorter, and much more readable.

http://www.journals.plos.org/plosone/s/file?id=wjVg/PLOSOne_formatting_sample_main_body.pdf andhttp://www.journals.plos.org/plosone/s/file?id=ba62/PLOSOne_formatting_sample_title_authors_affiliations.pdf

Done

2. Please provide additional details regarding participant consent. In the ethics statement in the Methods and online submission information, please ensure that you have specified whether consent was written or verbal/oral. If consent was verbal/oral, please specify: a) whether the ethics committee approved the verbal/oral consent procedure, b) why written consent could not be obtained, and c) how verbal/oral consent was recorded. If your study included minors, please state whether you obtained consent from parents or guardians in these cases. If the need for consent was waived by the ethics committee, please include this information.

Done A paragraph was added in page 8 of the revised manuscript.” Each participant gave informed written consent for the study, which was approved by the Ethical Committee of Regional Clinical Research of the Principality of Asturias.”

3. Please note that all PLOS journals ask authors to adhere to our policies for sharing of data and materials:https://journals.plos.org/plosone/s/data-availability. According to PLOS ONE’s Data Availability policy, we require that the minimal dataset underlying results reported in the submission must be made immediately and freely available at the time of publication. As such, please remove any instances of 'unpublished data' or 'data not shown' in your manuscript and replace these with either the relevant data (in the form of additional figures, tables or descriptive text, as appropriate), a citation to where the data can be found, or remove altogether any statements supported by data not presented in the manuscript."

Done. All the not shown data were added to the text of the revised version of the manuscript .All the data of coagulation were added to page 16 of the revised manuscript.

4. We noticed you have some minor occurrence(s) of overlapping text with the following previous publication(s), which needs to be addressed:

https://doi.org/10.1016/j.molimm.2007.10.013

https://doi.org/10.1093/infdis/jit158

https://doi.org/10.1097/GIM.0b013e318039b23d

https://doi.org/10.1016/j.molimm.2007.10.013

Done. All the overlapping text with previous publications, mostly from our group, has been written again.

In your revision ensure you cite all your sources (including your own works), and quote or rephrase any duplicated text outside the Methods section. Further consideration is dependent on these concerns being addressed.

Done

5. Thank you for including your ethics statement:

"Each participant gave informed consent for the study, which was approved by the Ethics Committee of the HUCA.".

i) Please amend your current ethics statement to include the full name of the ethics committee/institutional review board(s) that approved your specific study.

Done. A paragraph was added in page 8 of the revised manuscript.” Each participant gave informed written consent for the study, which was approved by the Ethical Committee of Regional Clinical Research of the Principality of Asturias.”

ii) Once you have amended this/these statement(s) in the Methods section of the manuscript, please add the same text to the “Ethics Statement” field of the submission form (via “Edit Submission”).

Done

For additional information about PLOS ONE ethical requirements for human subjects research, please refer tohttp://journals.plos.org/plosone/s/submission-guidelines#loc-human-subjects-research.

6. Thank you for stating in your Funding Statement:"Partially funded by a Fondo de Investigaciones Sanitarias (FIS) grant PI16/01999 given to VA.T he funders had no role in study design, data collection and analysis, decision to publish, or preparation of the manuscript".

Done.A paragraph was added to page 22 ot the revised manuscript: “This study was partially funded by a Fondo de Investigaciones Sanitarias (FIS) grant PI16/01999 given to VA. The funders had no role in study design, data collection and analysis, decision to publish, or preparation of the manuscript . There was no additional external funding received for this study.”

Done. A paragraph was added to page 22 ot the revised manuscript: “This study was partially funded by a Fondo de Investigaciones Sanitarias (FIS) grant PI16/01999 given to VA. The funders had no role in study design, data collection and analysis, decision to publish, or preparation of the manuscript . There was no additional external funding received for this study.”

Done 

7. We note that you have included the phrase “data not shown” in your manuscript. Unfortunately, this does not meet our data sharing requirements. PLOS does not permit references to inaccessible data. We require that authors provide all relevant data within the paper, Supporting Information files, or in an acceptable, public repository. Please add a citation to support this phrase or upload the data that corresponds with these findings to a stable repository (such as Figshare or Dryad) and provide and URLs, DOIs, or accession numbers that may be used to access these data. Or, if the data are not a core part of the research being presented in your study, we ask that you remove the phrase that refers to these data.

Done .All the not shown data were added to the text of the revised version of the manuscript .All the data of coagulation were added to pages 16 of the revised manuscript..

Additional Editor Comments:

1- The authors should use a language-editing service to refine the use of English in their manuscript and submit an "Editing Certificate" with the revised version of the manuscript.

Done. Joshua Fierer, Professor of Medicine and Pathology of the University of California San Diego (UCSD) did a complete scientific and English language editing of the original manuscript. This is the reason we decided to include him among the authors. A separate statement from Prof. Fierer confirming his contribution to the work is attached to the cover letter to the Editor.

2- Abbreviations must be spelled out on first mention particularly in the abstract.

Done

3- Authors are advised to revise all the manuscript to ensure that all gene names are written in italic font.

Done.

4- Authors should pay attention to the use of "Gender" terminology and substitute by the "sex" terminology as it appears from the results they mean the normal biological differences between males and females.

Done although all mentions to sex differences among CTSG N125S SNP genotypes have deleted from the revised version of the manuscript as Reviewer # 1 recommends.

5- Thanks to the authors for providing the PCR primer sequences. Are these primers -designed or derived from other published work? 

The primers were derived from previous works (reference 5. Ludecke B, Poler W, Olek K, Bartholome K. Sequence variant of the human cathepsin G gene. Hum Genet 1993; 91.83-84.; reference 6. Hermann SM, Funke-Kaiser H, Schmidt-Petersen K, Nicaud V, Gautier-Bertrand M, et al. Characterization of polymorphic structure of cathepsin G. Role in cardiovascular and cerebrovascular diseases. Arterioscler Thromb Vasc Biol 2001, 21: 1538-1543.

If the former, please provide the name of the program you applied in its formal citation in the text, or provide the citations you follow if they were derived from other publications.

6- The PCR results need more elaboration and clarification (recommended for acceptance). The authors wrote that "PCR products were incubated overnight at 37º C with the restriction enzyme.

Done. The PCR products overnight incubation at 37ºC is correct. A complete paragraph on PCR assay was added to pages 8 and 9 of the revised version of the manuscript “Measurement of the N125S (rs45567233) genetic polymorphism of CTSG gen .Ten ml of blood from each patient and controls were collected in potassium-EDTA glass tubes. Genomic DNA was extracted from peripheral leukocytes with a salting-out method and stored at -70º C until use. The N125S (rs45567233) polymorphism of the CTSG was analyzed in patients and controls by PCR. The PCR primers used for amplifying the region of the gene containing the N125S polymorphism (rs45567233) were: Forward: 5’-6-FAM-GCTGAGCGGGAACGCCTACA-3’ and Reverse: 5´-GCTGAGCGGGAACGCCTACA-3’. These primers amplified a fragment of 263 bp. The amplification protocol consists on an initial denaturation at 94º C for 5 minutes, followed by 35 cycles of denaturation at 94º C for 1 minute, annealing at 61º C for 1 minute and elongation at 72º C for 1 minute, and a final extension at 72º C for 7 minutes. PCR products were incubated overnight at 37º C with the restriction enzyme SduI (Thermo Fisher Scientific Inc, Waltham, MA, USA). Results were determined by using the Peak ScannerTM Software v1.0 (Applied Biosystem, Foster City, CA, USA) following capillary electrophoresis on ABI PRISM® 3130xl Genetic Analyzer (Applied Biosystem, Foster City, CA, USA). The CTSG N125S polymorphism was also genotyped individually for each osteomyelitis patient and control. PCR products were electrophoresed on a 2% low-melting agarose gel, and the fragments were then excised from the gel, purified with spin columns (DNA gel extraction Kit; Millipore, Billerica, MA,USA), and the fragments were directly sequenced on an ABIPrism 310 Genetic Analyser (Applied Biosystems, FosterCity, CA, USA).(Fig.1)

 Results were determined by using the Peak ScannerTM Software v1.0". The authors should send a photo related to the original scanner and an edited one for publication (in which all the details of band size for each genotype are written clearly) to facilitate replication of the work by future readers.

Done. A new Figure 1 includes a photo of the gel (A), and of the original peak scan (B) in which the band size of genotypes GG, AG and AA are seen. This new Fig.1 also includes the sequencing information of each genotype (C).

7- The quality control measurements the authors followed either in their PCR or other laboratory works, including ELISA, etc. should be written in details.

Done

ANSWERS TO THE REVIEWERS 

Reviewer #1: 

The study concept is interesting and can be of clinical significance.

My main concern is that the authors recruited an accepted number of patients (329) and controls (415) for and when they made the correlation with the evaluated parameters, they included very small groups of individuals (10 to 27 per group). The validity of these results is an issue and the bias in the selection of these individuals could be another problem.

We started recruiting osteomyelitis (OM) patients for several genetic studies since January 1998. So far we have recruited 329 adult patients with OM and collected their DNA. A large population of OM patients and controls is needed for genetic studies. However for CTSG, LF, MMPs and sRANKL assessment assays much smaller sample sizes are needed. Therefore we recruited 27 OM patients with acute bone infection consecutively admitted to the HUCA between 2016-18. In addition, 415 healthy HUCA Blood Bank donors, matched for sex and age with the OM patients, were used as controls, of them 21 were used for CTSG, LF, MMPs and sRANKL assessment.

Other points include

The title should determine the polymorphism ID and the study ethnic group.

Done. The title has been changed to “The N125S polymorphism in the cathepsin G gene (rs45567233 )is associated with susceptibility to osteomyelitis in a Spanish population.”

-value is sometimes mentioned as "equals" and other occasions as "less than". It should be consistent.

Done. “Equals” was used for individual p values throughout the manuscript. However when several p values are aggregated “less than”was used.

Page 4 last line. The authors mentioned that no similar association studies has been done so far. The reference dated 2006.

Done. The sentence was modified to “No associations of the CTSG N125S polymorphism with infections have been reported �7�”

Page 5, last paragraph is a mixture of the hypothesis and information extracted from reference no.9. Please rewrite.

Done. The sentence mentioning the association of high lactoferrin parotid saliva levels with periodontitis was moved from the paragraph backwards in the Introduction section.

There are linguistic style and punctuation errors. The manuscript needs linguistic revision. For example a sentence is repeated page 7, 

Done. English language was edited throughout the revised manuscript by Prof.Joshua Fierer from UCSD, a native English speaker. The repeated sentence was deleted.

parenthesis in page 19,

Done

 the tense in discussion (mixture of present and past tenses), first line in the second paragraph p23, .....

Done

The authors provided some data analysis based on the gender. It would be better to remove that as the female group is a minority.

Done. All mentions to sex differences among CTSG N125S SNP genotypes have have deleted from the revised version of the manuscript and reference 23 from the original version of the paper as Reviewer # 1 recommends.

Table 2 is confusing. Please separate into two tables (one for and the other for frequency.

Done. Two new tables, Table 2 with the genotype frequencies and Table 3 with the allelic frequencies were introduced in the revised version of the manuscript. 

The use of Yates correction is not preferred by some . I am not sure if it is the best to use here.

Done. Although the �2 obtained by the Yates correction is more exigent than the the �2 obtained by the Mantel -Haenszel test we have included both statistical tests in the revised version of the manuscript. as Reviewer # 1 suggests.

Reviewer #2: The authors presented an interesting study evaluating the association between CTSG polymorphisms and the occurrence of and its clinical characteristics. The study is well-conducted, but I suggest the following edits:

* Abstract: The authors did not mention the objective of their study in the abstract.

Done. A new paragraph was added to the abstract: “The aim of this study was to analyze a potential association between a CTSG gene polymorphism (Asn125Ser or N125S, rs45567233), that modifies CTSG activity, and could affect susceptibility to, or outcome of, bacterial osteomyelitis.“

* Introduction: the authors should present the different alleles, in their study.

Done. A new paragraph was added to the Introduction section:” Single carriers of the variant G genotype are heterozygous (GA), double carriers are homozygous (GG) while those that do not carry the variant G allele are wild type and have the AA genotype”

* Methods: "We enrolled 329 adult patients with a diagnosis of bacterial between January 1998 and December 2018" Was this a part of a hospital database system or a one study effort that lasted 20 years?

We started recruiting osteomyelitis (OM) patients for genetic studies since January 1998. So far we have recruited 329 adult patients with OM and collected their DNA.A large population of OM patients and controls is needed for genetic studies. However for CTSG, LF, MMPs and sRANKL assessment much smaller sample sizes are needed. For this purpose we recruited 27 with acute bone infection admitted to the HUCA between 2016-18. In addition, 415 healthy HUCA Blood Bank donors, matched for sex and age with the OM patients, were used as controls, 21 of them were used for CTSG, LF, MMPs and sRANKL assessment.

.

- Also, was there a systematic sample size calculation?

A paragraph regarding sample size calculation was added to the revised version of the manuscript:” Sample size calculation : Power calculation using one control for each osteomyelitis patients indicated a need for 746 individuals, 373 osteomyelitis patients and 373 controls to have a power of 80% at a confidence interval of 90% to detect a prevalence of 17% of the CTSG N125S polymorphism carriage among osteomyelitis patients considering that this polymorphism has a known prevalence of 10% in the Caucasian European population “

- "was diagnosed using clinical and roentgenographic findings" Please clarify?

Done. A paragraph was added to clarify this point. We have followed the osteomyelitis diagnostic criteria of Lew and Waldvogel. (Lancet 2004; 364: 369-79, reference 10): “A diagnosis of osteomyelitis was made by clinical and roentgenographic criteria. Surgical and sinus tract pus samples were cultured in all the osteomyelitis patients to make an etiologic diagnosis. The presence of bone sequestra and/or sinus tract in bone X ray, CT or MRI, a positive Ga67 uptake bone scan, or a positive culture of the surgical bone sequestra or sinus tract were diagnostic of osteomyelitis �10].”

- The molecular techniques used in the current study are well-described.

* Results: The age and gender of control subjects and their comparisons to patients should be mentioned in the results' first paragraph. Because they are supposed to be matched, no significant difference should be noted.

Done. A new paragraph was added to the results section: “The control group was made of 295 male (71.1%) and 120 female donors, with a median age of 59.1 years (41–60). Although the controls were 2.9 years younger than the patients there was not a significant difference between the groups (p=0.14) ”

- No association of the CTSG N125S polymorphism with or type of bone infection (acute vs. ), source of infection or microorganism isolated was found" the authors should mention at least p for such associations!

Done. A paragraph explaining this point was added to the revised version of the manuscript.: “No association of the CTSG N125S polymorphism with type of bone infection (acute vs. chronic) (p=0.7), source of infection (hematogenous vs.non hematogenous, p=0.9), (pressure ulcers vs. non pressure ulcers, p=0.5) or microorganism isolated (S. aureus vs. Gram negative bacteria p=0.9) was found “

”

* General: The manuscript needs a professional editing service because there are several grammatical errors across the entire manuscript.

Done. English language was edited throughout the revised manuscript by Prof.Joshua Fierer from UCSD, a native English speaker.

- Data availability statement should be added to the manuscript after the conclusion explaining where the underlying data could be found.

All the data are already shown in the revised version of the manuscript.

---

## [Decision Letter · Decision Letter 1]

8 Oct 2019

The N125S polymorphism in the <gwmw class="ginger-module-highlighter-mistake-type-1" id="gwmw-15703449758395636918275">cathepsin</gwmw> G gene (rs45567233) is associated with susceptibility to <gwmw class="ginger-module-highlighter-mistake-type-1" id="gwmw-15703449758393413330348">osteomyelitis</gwmw> in a Spanish population.

PONE-D-19-18899R1

Dear Dr. Asensi,

We are pleased to inform you that your manuscript has been judged scientifically suitable for publication and will be formally accepted for publication once it complies with all outstanding technical requirements.

Shortly after the formal acceptance letter is sent, an invoice for payment will follow. To ensure an efficient production and billing process, please log <gwmw class="ginger-module-highlighter-mistake-type-1" id="gwmw-15703449930245554470188">into</gwmw> Editorial Manager at https://www.editorialmanager.com/pone/, click the "Update My Information" link at the top of the page, and update your user information. If you have any billing related questions, please contact our Author Billing department directly at authorbilling@plos.org.

With kind regards,

Manal S. <gwmw class="ginger-module-highlighter-mistake-type-1" id="gwmw-15703450150231143822811">Fawzy</gwmw>, Ph.D., M.D.

Academic Editor

PLOS ONE

Additional Editor Comments (optional):

The authors have adequately addressed the concerns raised by the editor and the reviewers. Thank you

Reviewers' comments:

Reviewer's Responses to Questions

**Comments to the Author**

1. If the authors have adequately addressed your comments raised in a previous round of review and you feel that this manuscript is now acceptable for publication, you may indicate that here to bypass the “Comments to the Author” section, enter your conflict of interest statement in the “Confidential to Editor” section, and submit your "Accept" recommendation.

Reviewer #2: All comments have been addressed

2. Is the manuscript technically sound, and do the data support the conclusions?

Reviewer #2: Yes

3. Has the statistical analysis been performed appropriately and rigorously? 

Reviewer #2: Yes

4. Have the authors made all data underlying the findings in their manuscript fully available?

The PLOS Data policy requires authors to make all data underlying the findings described in their manuscript fully available without restriction, with rare exception (please refer to the Data Availability Statement in the manuscript PDF file). The data should be provided as part of the manuscript or its supporting information, or deposited <gwmw class="ginger-module-highlighter-mistake-type-3" id="gwmw-15703450284479521618669">to</gwmw> a public repository. For example, in addition to summary statistics, the data points behind means, medians and variance measures should be available. If there are restrictions on publicly sharing data—e<gwmw class="ginger-module-highlighter-mistake-type-3" id="gwmw-15703450313334784379329">.</gwmw>g. <gwmw class="ginger-module-highlighter-mistake-type-1" id="gwmw-15703450319090844961165">participant</gwmw> privacy or use of data from a third party—those must be specified.

Reviewer #2: Yes

5. Is the manuscript presented in an intelligible fashion and written in standard English?

PLOS ONE does not copyedit accepted manuscripts, so the language in <gwmw class="ginger-module-highlighter-mistake-type-3" id="gwmw-15703450342040199955324">submitted</gwmw> articles must be clear, correct, and unambiguous. Any typographical or grammatical errors should be corrected at revision, so please note any specific errors here.

Reviewer #2: Yes

6. Review Comments to the Author

Please use the space provided to explain your answers to the questions above. You may also include additional comments for the author, including concerns about dual publication, research ethics, or publication ethics. <gwmw class="ginger-module-highlighter-mistake-type-3" id="gwmw-15703450378082981490053">(</gwmw>Please upload your review as an attachment if it exceeds 20,000 characters)

Reviewer #2: All my comments were adequately addressed. I have no further comments on this manuscript and I recommend it for publication.

7. PLOS authors have the option to publish the peer review history of their article (what does this mean?). If published, this will include your full peer review and any attached files.

If you choose “no”, your identity will remain <gwmw class="ginger-module-highlighter-mistake-type-3" id="gwmw-15703450417888275426159">anonymous but</gwmw> your review may still be made public.

Reviewer #2: No

<gdiv></gdiv>

---

## [Editor Report · Acceptance letter]

15 Oct 2019

PONE-D-19-18899R1 

The *N125S* polymorphism in the cathepsin G gene *(rs45567233)* is associated with susceptibility to osteomyelitis in a Spanish population. 

Dear Dr. Asensi:

I am pleased to inform you that your manuscript has been deemed suitable for publication in PLOS ONE. Congratulations! Your manuscript is now with our production department. 

With kind regards,

on behalf of

Professor Manal S. Fawzy 

Academic Editor

PLOS ONE